# Determining Irradiation Dose in Potato Tubers During Storage Using Reaction-Based Pattern Recognition Method

**DOI:** 10.3390/foods14244285

**Published:** 2025-12-12

**Authors:** Yana V. Zubritskaya, Anna V. Shik, Irina A. Stepanova, Sergey A. Zolotov, Polina Yu. Borshchegovskaya, Ulyana A. Bliznyuk, Irina A. Ananieva, Alexander P. Chernyaev, Igor A. Rodin, Mikhail K. Beklemishev

**Affiliations:** 1Skobeltsyn Institute of Nuclear Physics, Lomonosov Moscow State University, GSP-1, 1-2 Leninskiye Gory, 119991 Moscow, Russia; ignobillium@yandex.ru (S.A.Z.); alexeevapo@mail.ru (P.Y.B.); uabliznyuk@gmail.com (U.A.B.);; 2Department of Physics, Lomonosov Moscow State University, GSP-1, 1-2 Leninskiye Gory, 119991 Moscow, Russia; 3Department of Chemistry, Lomonosov Moscow State University, GSP-1, 1-3 Leninskiye Gory, 119991 Moscow, Russia; shik.1966@mail.ru (A.V.S.); irishan@mail.ru (I.A.A.);; 4Lomonosov Institute of Fine Chemical Technologies, MIREA—Russian Technological University, 78 Vernadsky Ave., 119571 Moscow, Russia

**Keywords:** food irradiation, dose determination, potatoes, indicator reaction, photometry, fluorimetry, chemometrics

## Abstract

Food irradiation is increasingly used to extend shelf life and control pests and diseases. Monitoring post-treatment doses typically relies on expensive, laborious instruments and may miss low doses. We previously proposed a chemical fingerprinting method that estimates dose based on indicator reaction rates, but this approach was tested only on freshly irradiated samples. In this study, we investigated the feasibility of determining the order of magnitude of dose in irradiated raw potato tubers after several days of storage. A completely randomized experimental design was used. Water extracts of potatoes were assayed in oxidation–reduction and aggregation reactions in 96-well plates; reaction rates were tracked by absorbance and fluorescence and analyzed chemometrically. We could distinguish dose orders of magnitude (0, 100, 1000 Gy) after 0, 2, and 6 days of storage at 4 °C. The accuracy of dose recognition on day 6 was at least 97% by using SoftMax regression (SR) or linear discriminant analysis (LDA); irradiated and non-irradiated samples were confidently distinguished using partial least square–discriminant analysis (PLS-DA). The reaction-based method of dose assessment is simple, rapid, and does not require sophisticated equipment.

## 1. Introduction

Over the past decades food irradiation has become a technology of choice for industrial facilities worldwide since it is not only environmentally friendly but also highly adaptable to various needs of the agricultural industry [1]. It is not uncommon that some producers of agricultural goods fail to mark their products as irradiated, which creates a risk of repeated exposure to the foods that were not exposed to irradiation. In response to the food market demand catering to consumer rights protection it is important to develop a methodology for identification of irradiated food products of different categories in order to ensure food quality and safety. With the ongoing expansion of food irradiation goals due to high efficiency of this technology it is necessary to have a reliable approach applicable to a wide range of products and irradiation goals [2,3,4]. Since relatively low doses, which are commonly used for surface irradiation of agricultural products, are hard to differentiate, our research focuses on developing a methodology applicable for determining relatively low irradiation doses absorbed by agricultural products immediately after irradiation and during storage.

While meat and fish products require deeper penetration of irradiation since the main purpose of irradiation is to reduce pathogens in the entire volume of the product, roots and crops call for low-energy electrons or photons to treat surface layers of products where most phytopathogens and sprouts can be found [5]. Potatoes are irradiated to inhibit phytopathogens, such as *R. Solani*, *Alternaria* and *Fusarium*, which cause major post-harvest losses, and inhibit sprouting to prolong potato shelf life. To ensure high irradiation efficiency in potato, which have complex geometry, it is important to enable maximum irradiation dose in the surface layers of potatoes and control dose distribution in potato tubers during irradiation [6]. Since low-energy electrons and photons target the surface layers of the potato tubers with the maximum irradiation dose observed in the layers containing pathogens and sprouts, low-energy irradiation appears to be the most effective irradiation method for potato processing.

There are multiple processes occurring in irradiated food products. For instance, in potato tubers, weight loss, a much lower concentration of ascorbic acid and higher concentrations of sucrose, glucose and fructose are observed as a result of irradiation [7,8]. Irradiation leads to the destruction of glycosidic bonds and oxidative destruction of polysaccharides, formation of carboxyl and carbonyl groups, and a decrease in the viscosity of starch due to its depolymerization [9,10,11]. Increased starch hydrolysis, in turn, can lead to a short-term increase in the intensity of pulp respiration. Concentration of some amino acids can increase during the first day after irradiation, while concentration of other amino acids can decrease [12]. Changes in the chemical composition of the irradiated product may be the result of not only direct exposure to ionizing radiation but also secondary reactions with reactive oxygen species [13]. In particular, molecular oxygen, when irradiated, forms ozone, which is capable of attacking a carbon atom in a polysaccharide chain, breaking glycosidic bonds, and also abstracting a hydrogen atom from a carbon atom in a glucose chain to form a hydroxyl radical, which damages carbohydrates, proteins, lipids, and amino acids. The reactive oxygen species (ROS) OH^•^, $H_{2}O^{+•}$, $O_{2}^{-•}$ formed during irradiation also interact with polysaccharides, forming alkyl radicals, which then transform into carbonyl and carboxyl compounds [14]. ROS are generated even at low doses (50–150 Gy) used for extending the storage life of potatoes primarily by inhibiting sprouting. ROS can cause breaks in the DNA of rapidly dividing cells, preventing the meristematic tissue in the tuber eyes from receiving the genetic information needed for cell division and growth, thereby stopping sprouting. Low-dose irradiation can also affect the activity of enzymes involved in carbohydrate metabolism; if altered, it can suppress the sugar accumulation that typically occurs during sprouting [15]. Irradiation at doses significantly higher than those needed for sprout inhibition can cause enzyme inactivation, cell wall damage, production of volatile compounds, which can produce off-flavors. Doses over 150 Gy can harm the potato’s natural immunity, making it more vulnerable to disease and pathogens, which defeats the purpose of extending shelf-life [16]. The formation of new compounds as a result of irradiation can be dose-dependent, and the determination of the absorbed dose is possible using chemical analysis methods.

A number of methods are used to identify the fact of irradiation and determine the absorbed dose, each with its own limitations. For example, the electron spin resonance (ESR) is not applicable to products that do not contain solid particles such as cellulose, calcium, or bone; the same pertains to thermoluminescence [17]. The photostimulated luminescence technique places high demands on the storage and illumination conditions of the samples [18]. The use of GC-MS is based on the determination of trace volatile compounds such as alcohols, aldehydes, ketones [19], tyrosine isomers, creatine, trimethylamine oxide, and β-sitosterol [20]; these methods are efficient but have low sample throughput and require expensive instruments. Tests based on measuring antioxidant capacity, such as 2,2-diphenyl-1-picrylhydrazyl (DPPH) and ferric reducing antioxidant power (FRAP) assays, are mostly applicable for doses above 1 kGy [21]. Methods based on the determination of specific markers may not be applicable to all foods; for example, tests for 2-alkylcyclobutanones (ACB) work for products with a fat content of more than 1%, and the test for dihydrothymidine requires isolation and purification of DNA [22]. Most of the mentioned methods rely on stationary instruments and cannot be implemented outside the laboratory. The disadvantages of the methods also include complicated sample pretreatment, long analysis time, and low accuracy in determining small doses [20] that are necessary, for example, to stimulate potato sprouting [23].

For dose estimation in potatoes, most popular were electrochemical and thermoluminescence techniques. To determine a dose of 100 Gy, impedance ratio measurements were used; non-irradiated tubers of the same variety were used as controls [24]. A dielectric method was proposed to assay 50–200 Gy doses in potatoes [25]. Thermoluminescence measurements using mineral particles adhered to tubers were applied for dose detection in potatoes [26,27].

A different group of methods for dose assessment includes fingerprinting techniques: electronic nose and electronic tongue, which are based on the detection of volatile and soluble compounds, respectively, without determining the concentrations of specific markers [28]. Besides dose control, fingerprinting methods are widely used to distinguish between samples with similar composition: detect counterfeits [29], identify manufacturers [30] or pollution sources [31], perform medical diagnostics [32]. We have developed a reaction-based fingerprinting strategy that allows discrimination between samples using indicator reaction profiles [33,34,35,36]. In an effort to develop a universal and easy-to-implement dose assay technique, we applied this strategy to irradiated samples based on redox- and aggregation-type indicator reactions [34,35,36]. The components of the sample formed during irradiation interact with intermediate active particles generated in the course of the reactions, which entails different shapes of kinetic curves and allows us to distinguish between the dose values with an accuracy of an order of magnitude. The resulting optical signals (light absorption and fluorescence intensity) are processed using chemometric methods. A distinctive feature of this approach is the use of time-dependent processes, which, unlike “static” methods, provide more information about the samples and allow for more effective detection of subtle differences in their composition. Kinetic-based optical fingerprinting methods are fast, inexpensive, and versatile; they do not require full-spectrum or other complex equipment other than photo cameras and light sources and do not require highly skilled operators.

The majority of shelf-life studies of irradiated potatoes focus on the effect of irradiation on the stability of tubers [37]. The number of publications on the study of the chemical composition of irradiated tubers during storage is limited [38]. Although doses can be determined in potato tubers irradiated 3 days before analysis [24] or even after 20–25 days of storage [25], most studies on determining the irradiation dose in tubers are performed on the day of irradiation and do not examine the dependence of the result on storage time. Our previous studies of dose estimation methods in foods were also performed on freshly irradiated samples [34,35,36]. The purpose of this study was to evaluate the efficiency of the reaction-based fingerprinting strategy for detecting doses in raw potatoes during several days after irradiation. The experimental design and the principle of the kinetic fingerprinting method used are shown in Figure 1.

## 2. Materials and Methods

### 2.1. Object of Study and Reagents

Potato tubers of the Lina and Agata varieties, 4–5 cm in maximum size, were provided by the Siberian Federal Scientific Center for Agrobiotechnology of the Russian Academy of Sciences (SFNCA RAS). The Lina variety is a medium–early drought-resistant variety with a vegetation period of 75–90 days. The tubers with a starch content of 11.2–18.5% have a yellow and smooth skin and white flesh. The Agata variety is a medium–ripened variety with a vegetation period of 80–85 days. Tubers with a starch content of 12–14% have a yellow and dense skin and white flesh. Both varieties are high-yielding with high shelf life.

Carbocyanine dyes were synthesized by authors (dye **1** [39], dye **2** [40]). Ethanol 95% was procured from Bryntsalov-A (Moscow, Russia). Hydrochloric acid (conc.), hydrogen peroxide 30%, CuSO_4_·5H_2_O, Na_2_HPO_4_·2H_2_O, and KH_2_PO_4_ were obtained from Chimmed (Moscow, Russia). Cetyltrimethylammonium bromide was from Macklin (Shanghai, China), and ascorbic acid was from Meligen (Shcheglovo, Russia). Solutions were prepared in water obtained from a Millipore water purification system.

### 2.2. X-Ray Irradiation

The experiment was conducted on two Lina tubers and two Agata tubers. Nine parallelepipeds measuring 15 × 5 × 5 mm^3^ and weighing 0.5 g, eighteen pieces of each variety, were cut from the center of each potato tuber. Twelve pieces of each variety were placed in a 2-mL × 9-mm polypropylene tube for X-ray irradiation at the doses of 100 and 1000 Gy. Six pieces from three potato tubers of each variety were left as controls.

Irradiation was performed at the Physics Department of Moscow State University using a DRON UM-2 unit with a PUR5/50 power supply and BSV-23 X-ray tube with copper anode (Physics Department at Moscow State University, Moscow, Russia). The tubes were placed directly in front of a beryllium window of X-ray tube operating at 26 mA and 30 kV. The tubes were irradiated from two opposite sides, either side exposed to an equal dose amounting to half of the doses applied.

### 2.3. Dosimetry Control

The dose rate absorbed by the potato samples during X-ray irradiation was estimated using Fricke solution prepared using Mohr’s salt (NH_4_)_2_ Fe(SO_4_)_2_·6H_2_O (99% purity), sodium chloride (99.5% purity), and H_2_SO_4_ (95–99% purity) and stored in the dark under ambient temperature of 20 °C for 24 h. After the preparation, 0.5 mL of Fricke solution was placed in 2-mL polypropylene tubes and exposed to X-ray irradiation. Since the density of Fricke’s solution is close to the density of potato samples, it can be assumed that the dose rates absorbed by the dosimetry solution coincide with the dose rates absorbed by the potato samples. The dose absorbed by the solution was directly proportional to its absorbance measured on a spectrophotometer UV-3000 (TM Ecoview, “Promecolab”, St. Petersburg, Russia) at 304 nm, which increased linearly with the time of exposure as a result of the oxidation of ferrous ions (Fe^2+^) to ferric ions (Fe^3+^) caused by irradiation [41]. The calculation of the dose absorbed by Fricke solution was made using the formula:$$D=\frac{k∆S\left({Fe}^{3+}\right)}{\rho l\varepsilon G\left({Fe}^{3+}\right)},$$
where *k* = 9.65 × 10^6^ is a dimensionless coefficient, ∆S is the change in absorbance, *ε* = 2160 L·mol^−1^·cm^−1^ is the molar extinction coefficient of Fe^3+^ ions, *l* = 1 cm is the length of the optical path in the spectrophotometer cuvette, *ρ* = 1.024 g·cm^−3^ is the density of the dosimetry solution, and *G*(Fe^3+^) = 14.4 ions/100 eV is radiation-chemical yield for photons. Figure 1a shows the dependency of the dose absorbed by the dosimetry solution on irradiation time. The Fricke solution was irradiated with (1.1 ± 0.1) Gy/s.

To ensure uniform irradiation, each side of the potato samples was exposed to irradiation during an equal amount of time (Table 1). Nine identical samples of each variety were irradiated with each dose.

### 2.4. Dose Uniformity Control

Irradiation uniformity in potato samples was simulated using GEANT4-11.2.1 software, which is based on the Monte Carlo method [42]. Potato samples were represented as water parallelepipeds with the linear dimensions corresponding to the potato samples and irradiated with 10^6^ photons having energy spectrum shown in Figure 1b.

The irradiation method used in the computer simulation was consistent with the irradiation method used in the experiment (Figure 1c). To obtain dose distribution, water parallelepiped was divided into 150 × 50 × 50 cubic cells with an edge of 0.1 mm, and the dose absorbed by each cell was calculated as:$$D_{i}=\;\frac{\sum_{j=1}^{N_{i}}E_{ij}}{m_{i}},$$
where $\sum_{j=1}^{\;N_{i}}E_{ij}$ is the sum of the energy received by the *i*-th cell during *N_i_* interactions of photons with water, and *m_i_* is the cell mass. The margin of error in the dose was calculated using the formula:$$S_{i}=\;\sqrt{\left(\frac{1}{N_{i}-1}\sum_{j=1}^{N_{i}}E_{ij}^{2}-{\left(\frac{1}{N_{i}-1}\sum_{j=1}^{N_{i}}E_{ij}\right)}^{2}\right)},$$
where $\sum_{j=1}^{N_{i}}E_{ij}^{2}$ is the sum of squares of the energy received by the *i*-th cell for N_i_ interaction acts. The dose in each cell of the water parallelepiped was divided by the maximum dose value recorded in the water parallelepiped, and then the relative dose value in each cell was color-coded to show the dose distribution. The 3D-dose distribution map shown in Figure 1d was built for two-side irradiation of the water parallelepiped simulating the potato sample. As it can be seen from Figure 1d, the dose uniformity in water parallelepiped was 0.2.

However, such dose distribution did not affect the subsequent chemical stage of the experiment, since each of all potato parallelepipeds was used for making the potato extracts. For each irradiation dose, potato extracts were prepared using nine pieces from two different potato tubers.

After irradiation, the samples were stored at 4 °C, and fingerprinting analysis was performed on the day of irradiation and on the 2nd and 6th days after irradiation in order to estimate the possibility of discrimination of potato samples irradiated with different doses during a few days after irradiation. Two samples from two different potato tubers of each variety, irradiated with each dose (0, 100, and 1000 Gy), were examined on each day to estimate the accuracy of discrimination.

### 2.5. Sample Preparation

To prepare the samples, test tubes containing potatoes were filled with 940 µL of distilled water and 20 µL of a 0.1 mol/L ascorbic acid solution, which partially prevented the potato pieces from browning as a result of oxidation. The test tubes were then placed in an orbital shaker for 12 h at 120 rpm to obtain an extract for further storage in the refrigerator for 2 or 6 days of the experiment. The potato extracts were analyzed using redox and aggregation indicator reactions. Five parallel experiments were conducted for each sample, filling five wells of a 96-well plate (Thermo Scientific Nunc F96 MicroWell, white, cat. No. 136101 from Thermo Fisher Scientific, Waltham, MA, USA).

To conduct the redox reaction, the following solutions were added to the wells:(1)Phosphate buffer solution, 0.067 M in phosphate, pH 7.4, 30 µL;(2)Cetyltrimethylammonium bromide (CTAB), 0.001 M, 30 µL;(3)Distilled water, 125 µL;(4)Potato extract, 25 µL;(5)CuSO_4_ solution, 0.001 M, 30 µL;(6)H_2_O_2_, 1 M, 30 µL;(7)Dye **2** (Figure 2), 0.1 g/L, 30 µL.

The dyes were first dissolved in 95% ethanol to obtain a 1 g/L solution, which was stored at 4 °C. On the day of the experiment, this solution was diluted with water to obtain a 0.1 g/L solution in water–ethanol medium (9:1 *v*/*v*).

To perform the aggregation reaction, the following solutions were added to the well:(1)HCl, 0.1 M, 30 µL;(2)Distilled water, 145 µL;(3)CTAB, 0.001 M, 70 µL;(4)Potato extract, 25 µL;(5)Dye 1, concentration 0.025 g/L, 30 µL.

The time of reaction start was determined by adding the dye to the wells. After mixing, the absorbance/reflectance of the reaction mixtures and their fluorescence were periodically recorded using excitation at wavelengths of 254, 366, and 660 nm. To obtain visible-light images, the plate was photographed in a Visualizer 2 (Camag, Muttenz, Switzerland), and for near-infrared emission, a homemade NIR visualizer was used, which included red LEDs (660 nm) as a light source and a Nikon D80 camera with a long-pass light filter, cutting off visible light up to 700 nm [39]. For the aggregation reaction, imaging was performed once, and for the redox reaction, four times (every 10 min, counting from the moment the dye was added). Examples of the resulting images are shown in Figure 2.

The images were digitized using ImageJ software v. 2 to obtain color intensity values for the reaction mixtures. RGB splitting was performed for the visible absorption/reflection images. The results of photographic digitization were presented in the form of data tables, in which the rows corresponded to the samples and their replicates (that are called below ‘observations’). Each dose (0, 100, or 1000 Gy) was represented by two physical samples, and each sample was analyzed in five replicates (five wells of the plate) to give a total of 30 observations. The columns corresponded to the images obtained in two indicator reactions: for the aggregation reaction, R, G and B channels of visible absorbance/reflectance were used (3 columns), along with fluorescence intensities excited at 254, 366 and 660 nm (3 columns). For the redox reaction, total fluorescence intensities for emission excited at 366 and 660 nm (2 wavelengths × 4 timepoints = 8 columns) and for absorbance/reflectance, R, G and B channels at 4 timepoints (12 columns) were processed, which gave a total of 26 data columns. An example of a data table is provided in Appendix A. Separate data tables were compiled for each potato variety (two varieties in total) and for each day of the experiment (0, 2 and 6). The obtained data were analyzed using the Principal Component Analysis (PCA) and Linear Discriminant Analysis (LDA) methods using the XLSTAT Excel add-in, v2016.02.28451 (Lumivero, Denver, CO, USA), Partial Least Squares-Discriminant Analysis (PLS-DA) using Unscrambler X, v. 10.4 (Camo Software/AspenTech, Houston, TX, USA), and LDA and SoftMax Regression (SR, an analog of logistic regression) using dedicated software composed by the authors based on Python (v. 3.12) libraries. The results were visualized as two-dimensional graphs in the coordinates of the principal components PC1–PC2 (for PCA) or factors F1–F2; ellipses in the graphs were drawn for 90% confidence level.

For LDA, SR, and PLS-DA, the data were divided into training and validation sets. Random cross-validation was used to assess the model quality; the validation set included 5 observations out of 30 (17% of all observations in the data table). To assess the feasibility of dose recognition under real-world conditions, *k*-fold (“leave-one-sample-out”) cross-validation was used, in which whole samples (5 observations belonging to one sample) were sent to the validation set. During the LDA and SR validation process, the program assigned validation set observations to one of three classes (0, 100, and 100 Gy) by calculating the Mahalanobis distance between a point (observation) and a group of points of the corresponding class; an observation was assigned to the class with the minimal distance.

The dose discrimination accuracy *A* (%) was calculated as the number of correctly assigned observations divided by the total number of observations using the formula: *A* = *corr*/*all*·100%, where *corr* is the number of correctly assigned observations of the validation set, and *all* is the total number of observations in this set. The accuracy values presented in this paper were averaged over all observations in the validation set; their standard deviations ranged from 7% to 24% (0% for a value of 100%).

## 3. Results and Discussion

### 3.1. Selection of the Indicator System

Fingerprint methods do not reveal the nature of marker compounds that are responsible for the analytical signal; rather, they represent a “black box” methodology that does not interpret the signals: fine differences in composition result in signal differences that enable the samples to be distinguished. In this work we follow the reaction-based fingerprinting strategy that is based on indicator reactions conducted in the presence of samples.

Our previous data [34] showed the efficiency of some redox and aggregation-type indicator reactions in discriminating between potato doses. Accordingly, two reactions were chosen for this study: the oxidation of dye **1** by hydrogen peroxide, catalyzed by Cu^2+^ ions, to form colored and fluorescent products; a submicellar amount of surfactant (CTAB) was added to the solution to enhance the fluorescence of the emitting particles, which are poorly soluble in water. The aggregation-type reaction occurred between the anionic components of the sample and CTAB; during the formation of such ion pairs, dye **2** could enter the hydrophobic domains of the resulting nanoparticles with fluorescence enhancement. This mechanism was considered in more detail previously [39]. The signal in the aggregation-type reaction is virtually independent of time, for which reason it was measured only once. In contrast, the redox reaction proceeded over time, and four photographs of the plate were taken over 30 min (every 10 min). To construct a data table for chemometric processing, the signals obtained in both reactions were pooled. The general scheme of the experiment is shown in Figure 1.

### 3.2. Raw Data and Their Unsupervised Chemometric Processing

Kinetic curves obtained by photographing the reaction mixtures for half an hour vary in shape for different samples, changing with the potato variety, dose, and day after irradiation (Figure 3). The complex shape of the curves and their change with storage time (for example, the signal for the Lina variety decreases with time on day 0 and increases with time on days 2 and 6) may be due to interconversions of organic compounds occurring in the irradiated tubers during storage [6]. The complexity of the profiles may be caused by parallel degradation of the dye into several products, fluorescent and non-fluorescent, as a result of interaction with the oxidant. The rates of these processes may also depend on the composition of the sample, which is dose-dependent.

The raw signal–time profiles (Figure 3) hardly allow for distinguishing the doses. To simplify the picture and represent each kinetic curve as a single point on the score plot, we processed the data using principal component analysis (PCA), an unsupervised technique that provides an objective presentation of the data. Using PCA score plots, irradiated Agatha potato samples can be distinguished from non-irradiated ones (Figure 4), while the doses of 100 and 1000 Gy are inseparable by this method. To improve discrimination, supervised methods, SoftMax Regression (SR), Linear Discriminant Analysis (LDA) and Partial Least Squares–Discriminant Analysis (PLS-DA) were further used.

### 3.3. Supervised Chemometric Processing

Linear Discriminant Analysis (LDA) is a supervised machine learning technique that finds a linear combination of features that best separates classes by maximizing the ratio of between-class variance to within-class variance [43]. Applying LDA to the data allowed us to construct the score plots (Figure 5) demonstrating the ability to discriminate samples by dose for all storage periods and both potato varieties. Quantitative information is presented in Table 2, which lists the dose recognition accuracies. Accuracy values of 100% were considered excellent, 90–99% were considered good, and 80–89% were considered satisfactory.

The left-hand side of Table 2 (both indicator reactions used) shows that the doses in both varieties can be distinguished with good accuracy regardless of storage period, with the exception of day 2 for Lina variety (satisfactory accuracy). However, when only the redox reaction data were processed (the right-hand side of Table 2), the accuracy values were significantly lower, confirming the need to use both reactions to achieve discrimination.

Among the two chemometric techniques used for dose estimation (LDA and SR), SoftMax regression was found to be more effective (higher accuracy values than LDA, as shown in Table 2). SoftMax regression, or multinomial logistic regression, is a generalization of logistic regression for multi-class classification problems. It works by applying the so-called SoftMax activation function to convert scores for each class into probabilities, allowing the model to predict the probability of an input belonging to one of several possible outcomes [44]. SoftMax regression can outperform LDA on datasets where the classes are not normally distributed or do not have equal variance-covariance matrices. LDA relies on these assumptions, so its performance degrades when they are violated, whereas SoftMax regression is more robust to these violations and is better for non-linear decision boundaries. Softmax is also preferred in high-dimensional datasets where LDA covariance matrix can be unstable [45].

The ability to differentiate between the doses revealed that the signal remained dose-discriminating for several days after treatment. This result suggests that exposure to ionizing radiation induces chemical changes in potatoes that are stable over time and that the compounds responsible for the signal remain stable during storage (they cannot belong, for example, to the class of peroxides, which are rapidly degraded enzymatically).

### 3.4. Distinguishing Between Irradiated and Non-Irradiated Samples

A separate task is recognizing the fact of radiation treatment itself, which is important, for instance, for detecting fraud or preventing repeated irradiation of the same product. Using our data, we attempted to demonstrate that this task can be accomplished not only immediately after irradiation of samples but also at least several days later. All data were divided into two classes: controls (not irradiated) and both doses (100 and 1000 Gy combined). The results (Table 3) show that, for the SR and LDA techniques, this task is easier than recognizing all doses, and the accuracy values are generally higher than those in Table 2.

Moreover, for this two-class discrimination task, another efficient chemometric technique such as Partial Least Squares Discriminant Analysis (PLS-DA) can be used. PLS-DA is a supervised multivariate method that projects high-dimensional data into a lower-dimensional space using latent variables that maximize separation between predefined groups. Similar to SR and LDA, and unlike unsupervised methods such as PCA, PLS-DA is aware of the class labels of the samples. The latent variables are chosen to maximize the covariance between predictor variables (e.g., received signals) and response variables (class labels, e.g., doses). PLS-DA typically operates on two classes of data, which allows using this technique to discriminate between non-irradiated and irradiated samples. As a result of calculation, a reference variable *y* was assigned to each observation, which was equal to 0 for non-irradiated samples and 1 for the irradiated ones. An observation was considered to be correctly classified as ‘non-irradiated’, if the predicted *y* value was less than 0.5, and as ‘irradiated’, if it exceeded 0.5. The discrimination accuracy was calculated as the number of correctly classified observations to the total number of observations in the validation set.

The results of applying PLS-DA to the data, as well as similar results for SR and LDA, are presented in Table 3 (left-hand part: ‘random cross-validation’). The recognition accuracy of PLS-DA is 100% for all six cases (both potato varieties and three days), indicating an excellent accuracy of the chemometric model in the differentiation of irradiated and non-irradiated samples for different storage periods.

### 3.5. Distinguishing Between Doses Using k-Fold Validation

The accuracy values presented above were obtained using random cross-validation, which is useful in verification of the correctness of the chemometric model but is not suitable for the recognition of samples in a real-world experiment. To simulate the recognition of unknowns, cross-validation should involve withdrawal of entire samples (rather than random individual observations) from the dataset and including them in the validation set. In this study, the sample contained 5 parallel runs (5 dataset observations), so groups of five consecutive observations were included in the validation set during *k*-fold validation. Accuracy was calculated as the ratio of the number of correctly assigned samples to the total number of samples in the validation set. In this study, we only had two physical samples per dose, and using *k*-fold validation, the model had limited capacity to learn (when one sample was sent to the validation set, the model was trained on the remaining single sample).

For the reasons stated, the accuracy values obtained using *k*-fold cross-validation (Table 4) are significantly lower than those received with random cross-validation (Table 2). Nevertheless, even under these realistic conditions, accuracy on day 6 did not fall below 97% using SR and LDA for all doses, confirming the feasibility of confidently determining the dose in an unknown sample to an order of magnitude after the specified storage period. In particular, to obtain these results, it was important to evaluate different chemometric techniques: for the Lina variety, the SR method proved to be more effective, while for the Agata variety, the LDA method performed better in *k*-fold cross-validation.

PLS-DA also allowed using *k*-fold cross-validation but only for a two-class analysis, i.e., to distinguish between irradiated and non-irradiated samples (right-hand part of Table 3). For the Agatha variety, all accuracy values equal 100%, while, for the Lina variety, they do not exceed 80% on days 0 and 2, although for day 2 higher accuracy is achieved using the SR technique (90%). A tendency is observed that the ability to differentiate doses is not only maintained during storage but may even improve over time. Similar to the three-class analysis using the LDA/SR techniques, excellent accuracy was achieved on day 6 using PLS-DA, confirming the possibility of confidently determining the dose after a storage period.

## 4. Conclusions

In this study, we investigated doses in raw potatoes at various times after X-ray irradiation. In addition to the previously demonstrated accurate determination of doses on the day of treatment, we revealed the feasibility of estimating the doses in tubers within six days after irradiation. The kinetic-based fingerprinting method that we use is a simple, affordable and rapid technique that does not require full-spectrum instruments or highly qualified personnel. The results showed that the dose estimation accuracy to an order of magnitude (100 or 1000 Gy or non-irradiated) was at least 97% even on the sixth day of storage. The ability to distinguish irradiated samples from non-irradiated ones on the 6th day was 100%.

These findings are important for the practical application of the proposed methodology, as dose control issues can arise over various storage periods. Indirectly, these results indicate that the chemical changes that are responsible for the signal are stable during storage. If future studies determine that dose analysis is feasible up to several months after irradiation, it will have great practical importance. Experiments with longer shelf-life periods are required to explore this possibility.

The kinetic-based fingerprinting method that we use is a simple, affordable and rapid technique that does not require full-spectrum instruments or highly qualified personnel. By using other indicator reactions, it can be applied not only to raw potatoes but also to other food products (beef, chicken [35,36]). However, these advantages come at the cost of handling a large number of solutions (wet chemistry operations), analyzing standard samples alongside unknowns, and limited precision (up to an order of magnitude). Therefore, the potential for further development of this method lies in the development of ready-to-use test systems (“just add the sample”), improving signal stability over time, and the discrimination of doses that vary slightly from one another.

Another prospect of developing the method lies in reducing the number of indicator reactions for dose recognition. In this study involving potato tubers, two reactions were used, while, for other foods, a single reaction was often sufficient to estimate all doses with an order-of-magnitude accuracy. Selecting the optimal chemometric method is also important: SoftMax regression proved to be the most effective in case of random validation, but work with more practically important *k*-fold cross-validation showed that linear discriminant analysis can be equally effective for some samples. PLS-DA was unrivaled in solving two-class recognition tasks. Therefore, future studies of reaction-based dose estimation should explore the application of the broadest possible range of chemometric methods.

## Figures and Tables

**Scheme 1 foods-14-04285-sch001:**
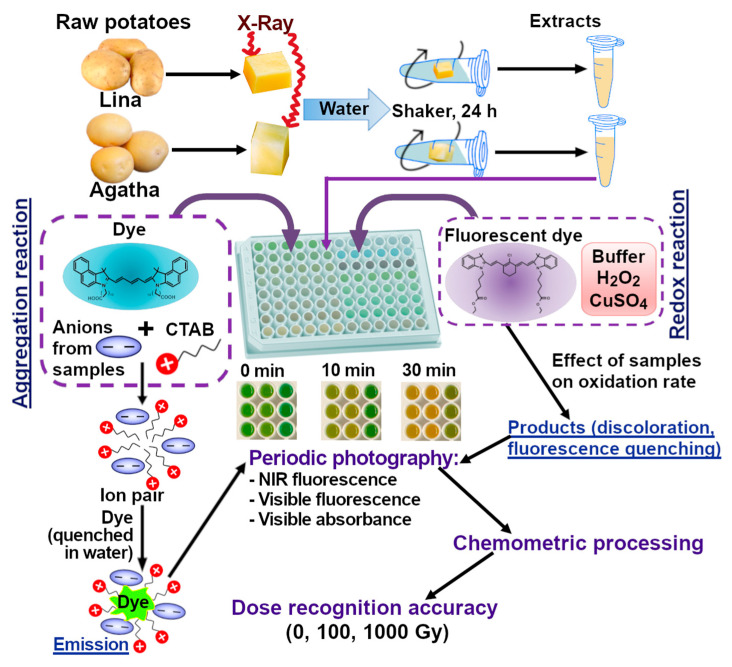
Main stages of the experiment (irradiation of potato tubers, extraction, indicator reactions and data processing).

**Figure 1 foods-14-04285-f001:**
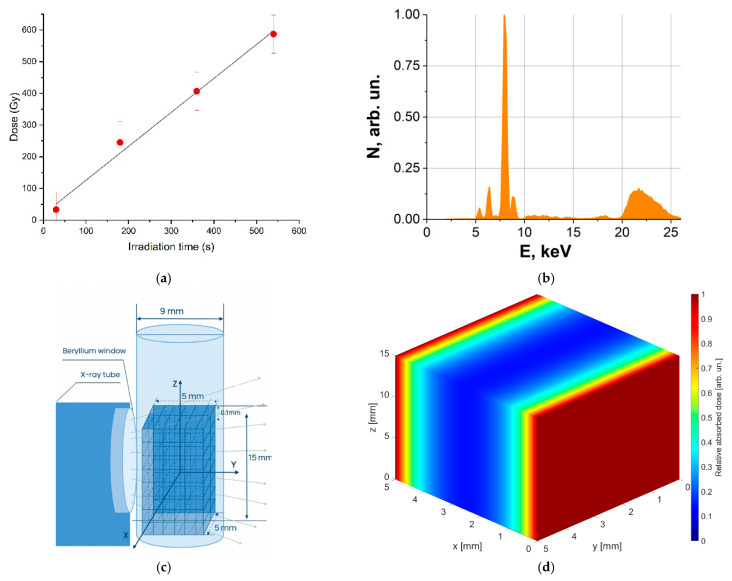
DRON UM-2 X-ray unit irradiation parameters: (**a**) Dependence of the dose absorbed by the Fricke solution on irradiation time. (**b**) The energy spectrum. (**c**) Simulation of bilateral irradiation of potatoes. (**d**) Relative dose distribution in a 5 mm thick, 5 mm long and 15 mm high water parallelepiped irradiated with X-rays.

**Scheme 2 foods-14-04285-sch002:**
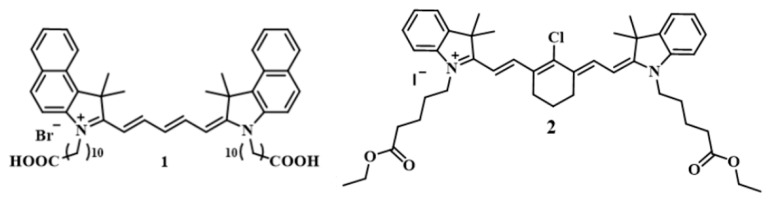
Structures of the dyes used for optical fingerprinting of potatoes.

**Figure 2 foods-14-04285-f002:**
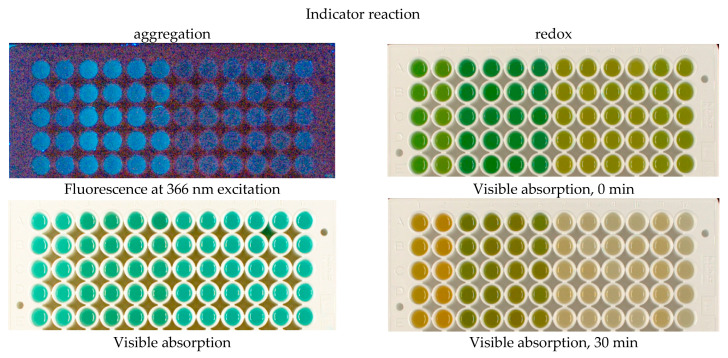
Examples of photographic images of the reaction mixtures. The indicator reactions were conducted with the samples of irradiated potato samples stored for 6 days. Potato extracts were presented as 2 samples per dose, 6 parallels per sample (wells in columns, from left to right in each photograph): Agata control-1, Agatha control-2, Agatha 100 Gy-1, Agatha 100 Gy-2, Agatha 1 kGy-1, Agatha 1 kGy-2, Lina control-1, Lina control-2, Lina 100 Gy-1, Lina 100 Gy-2, Lina 1 kGy-1, Lina 1 kGy-2.

**Figure 3 foods-14-04285-f003:**
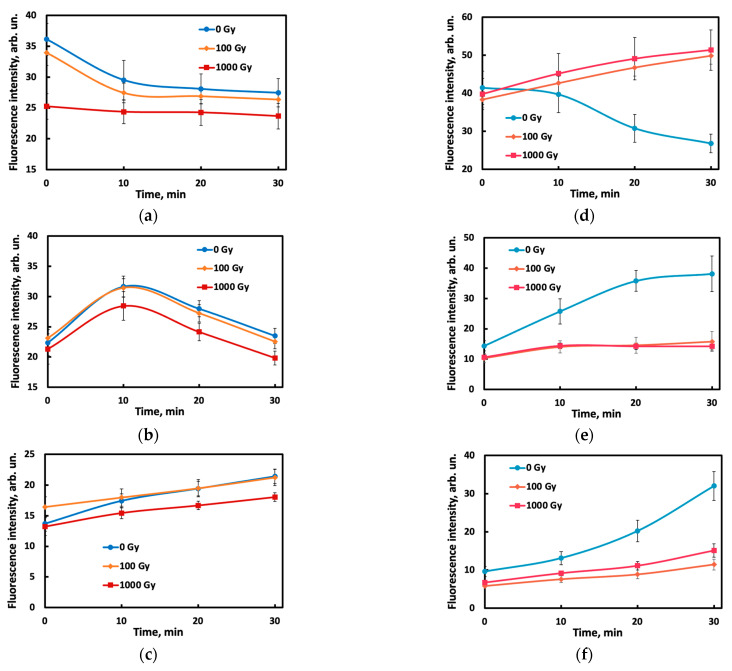
Time curves of the fluorescence intensity of the reaction mixtures in the near-IR range on the 0th (**a**,**d**), 2nd (**b**,**e**) and 6th (**c**,**f**) day for potato samples of the Lina (**a**–**c**) and Agata (**d**,**e**) varieties. (**a**) Lina, 0 days; (**b**) Lina, 2 days; (**c**) Lina, 6 days; (**d**) Agatha, 0 days; (**e**) Agatha, 2 days; (**f**) Agatha, 6 days.

**Figure 4 foods-14-04285-f004:**
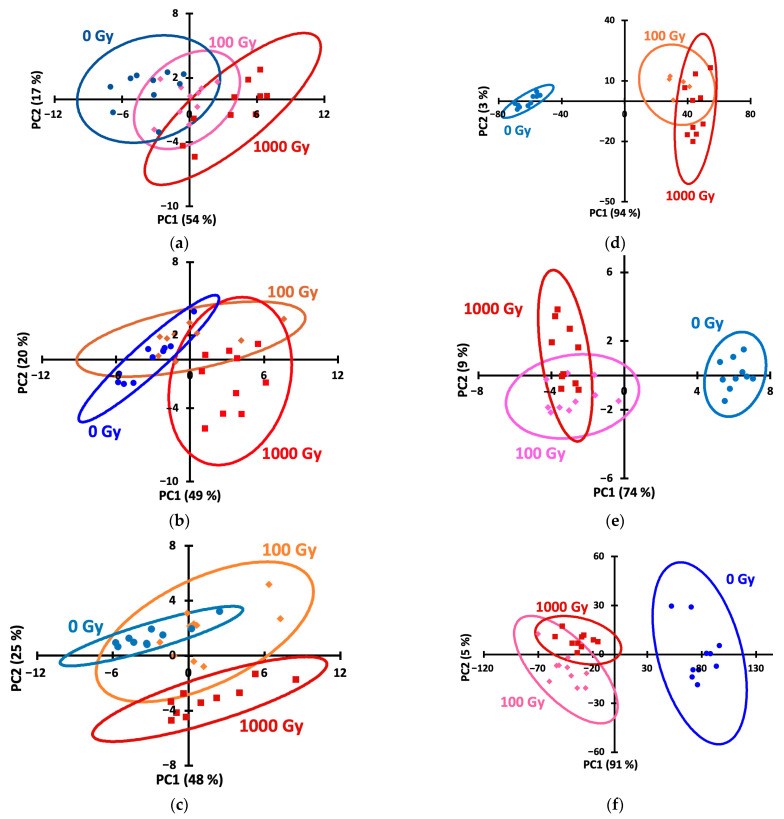
Principal component analysis (PCA) score plots for 0th (**a**,**d**), 2nd (**b**,**e**) and 6th (**c**,**f**) day for potato samples of the Lina (**a**–**c**) and Agata (**d**,**e**) varieties. (**a**) Lina, 0 days; (**b**) Lina, 2 days; (**c**) Lina, 6 days; (**d**) Agatha, 0 days; (**e**) Agatha, 2 days; (**f**) Agatha, 6 days.

**Figure 5 foods-14-04285-f005:**
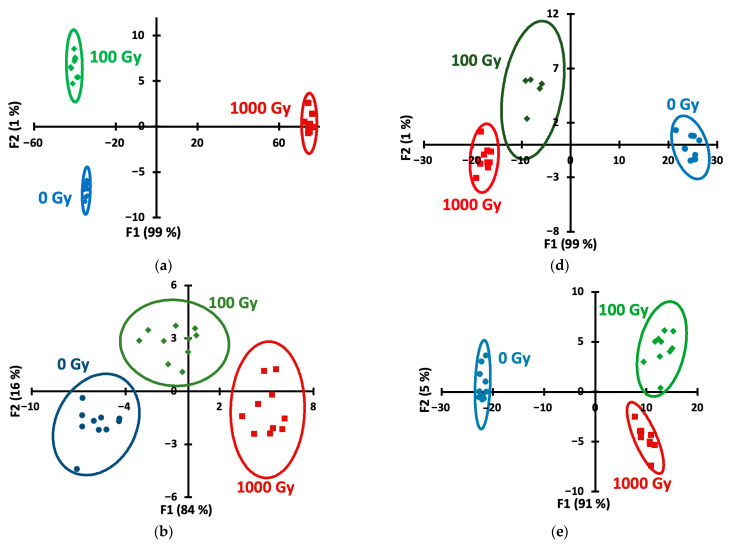
LDA score plots for days 0 (**a**,**d**), 2 (**b**,**e**) and 6 (**c**,**f**) for Lina (**a**–**c**) and Agatha (**d**–**f**) variety samples. (**a**) Lina, 0 days; (**b**) Lina, 2 days; (**c**) Lina, 6 days; (**d**) Agatha, 0 days; (**e**) Agatha, 2 days; (**f**) Agatha, 6 days.

**Table 1 foods-14-04285-t001:** Sample irradiation time and dose.

Sample	Irradiation Time, s	Calculated Dose, Gy
C_1_, …, C_6_	0	0
D_1-1_, …, D_1-6_	110	100
D_2-1_, …, D_2-6_	920	1000

**Table 2 foods-14-04285-t002:** Accuracy of recognition (%) of three doses (0, 100 and 1000 Gy) in potato tubers for various storage times, obtained by two chemometric techniques, SR and LDA.

Storage Day	Using Full Reaction Set (26 Data Columns)		Using Only the Redox Reaction (20 Data Columns)	
	SR	LDA	SR	LDA
Lina variety				
0	**93**	90	87	70
2	87	80	**93**	77
6	**97**	63	83	70
Agatha variety				
0	**100**	**100**	83	63
2	**93**	**96**	**93**	80
6	**100**	**97**	**97**	80

Note. Accuracy (%) was calculated as the ratio of the number of correctly assigned observations in the validation set to the total number of observations in it. Values exceeding 90% are shown in bold.

**Table 3 foods-14-04285-t003:** Discrimination accuracy (%) between non-irradiated and irradiated potato samples using two indicator reactions for various storage periods, obtained by three chemometric techniques.

Day of Storage	Cross-Validation Type					
	Random			*k*-Fold		
	SR	LDA	PLS-DA	SR	LDA	PLS-DA
Lina variety						
0	87	87	**100**	73	50	80
2	**97**	87	**100**	**90**	60	80
6	**97**	80	**100**	**90**	67	**100**
Agatha variety						
0	**100**	**100**	**100**	**100**	**100**	**100**
2	**100**	**100**	**100**	**100**	**100**	**100**
6	**100**	**93**	**100**	**100**	**100**	**100**

Note. Accuracy values greater than 90% are shown in bold.

**Table 4 foods-14-04285-t004:** Accuracy of recognition (%) of three doses (0, 100 and 1000 Gy) in potato tubers using full reaction set for various storage times obtained by k-fold cross-validation.

Storage Day	Chemometric Technique	
	SR	LDA
Lina variety		
0	60	43
2	63	53
6	**97**	57
Agatha variety		
0	60 *	**100** *
2	77	53
6	77	**97**

* Outliers in visible light photograph intensities for the Agatha 100-Gy sample on day 0 were excluded from the data. For this reason, no *k*-fold cross-validation was possible with this dose for day 0, and the accuracy values were calculated for 1 kGy and control tubers only. Accuracy values greater than 90% are shown in bold.

## Data Availability

The original contributions presented in this study are included in the article and Appendix A. Further inquiries can be directed to the corresponding authors.

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
