# Peer review of "Determining Irradiation Dose in Potato Tubers During Storage Using Reaction-Based Pattern Recognition Method"

_foods, 2025, doi:10.3390/foods14244285_

Round 1

Reviewer 1 Report

Comments and Suggestions for Authors

This manuscript presents a novel, reaction-based fingerprinting strategy for determining the magnitude of irradiation dose in raw potato tubers during storage. Here are some issues need to be addressed before the manuscript can be considered for publication.

  1. The abstract is too long, and contains a large amount of background descriptions.
  2. As described in Introduction part, many methods have developed for measuring the irradiation dose of foods, though they have many defects, but the authors also should not despite the advantages. Please add some discussion of the present work regarding whether it could achieve the advantages of those technologies.
  3. As shown in Method part, “Twelve pieces of each variety were placed in a 2-mL × 9-mm polypropylene tube for X-ray irradiation at the doses of 100 and 1000 Gy. Six pieces from three potato tubers of each variety were left as controls”. During the industrial production, the potato would not be treated with irradiation after cutting. Please explain why did authors design the experiment in this way?
  4. The study used only two tubers per variety and only two varieties. With only two physical samples per dose, the results based on model trained on a very limited dataset. The conclusions about the method's accuracy, especially for k-fold validation, might be not yet statistically robust. More sample and doses should be tested.
  5. Please explain why did authors just select X-ray irradiation at the doses of 100 and 1000 Gy? Why not 50, or 500 Gy? The storage time is also confusion.
  6. The fingerprinting strategy is "black box," what is the method actually sensing? Some information about the chemical changes induced by irradiation that are detected by the indicator reactions should be supplemented.
  7. How accurate is the present method? Can this method accurately determine the irradiation dose? And it might be largely influenced by the sampling or sample preparation methods. 
Comments on the Quality of English Language

none.

Author Response

Reviewer 1

  1. The abstract is too long, and contains a large amount of background descriptions.

Reply. We have shortened the Abstract from 216 to 185 words by removing a less important part of the information. The background descriptions are deemed important for the non-specialist chemist and have been left in.

  1. As described in Introduction part, many methods have developed for measuring the irradiation dose of foods, though they have many defects, but the authors also should not despite the advantages. Please add some discussion of the present work regarding whether it could achieve the advantages of those technologies.

Reply. We added more comparison with the existing technologies under Conclusions as follows:

The kinetic-based fingerprinting method that we use is a simple, affordable and rapid technique that does not require full-spectrum instruments or highly qualified personnel. However, these advantages come at the cost of handling a large number of solutions (wet chemistry operations), analyzing standard samples alongside unknowns, and limited precision (up to an order of magnitude). Therefore, the potential for further development of this method lies in the development of ready-to-use test systems ("just add the sample"), improving signal stability over time, and the discrimination of doses that vary slightly from one another.

  1. As shown in Method part, “Twelve pieces of each variety were placed in a 2-mL × 9-mm polypropylene tube for X-ray irradiation at the doses of 100 and 1000 Gy. Six pieces from three potato tubers of each variety were left as controls”. During the industrial production, the potato would not be treated with irradiation after cutting. Please explain why did authors design the experiment in this way?

Reply. Yes, industrial irradiation facilities can treat the whole tubers. However, the current trend is to switch to low-energy irradiation for surface treatment of agricultural products to make irradiation affordable to smaller businesses, so we used a compact 26 keV X-ray apparatus generating low-energy photons that are able to penetrate the depth of up to 5 mm. In our experiment, the samples have the dimensions and mass which, on the one hand, are sufficient for preparing a potato extract, and, on the other hand, ensure the irradiation of the entire volume of the samples. Whole tubers were irradiated by 1 MeV accelerated electrons in our paper published in Food Chem. in 2023; in that case, the upper layer of the tubers was peeled and extracted to estimate the dose. This work presents a different option.

  1. The study used only two tubers per variety and only two varieties. With only two physical samples per dose, the results based on model trained on a very limited dataset. The conclusions about the method's accuracy, especially for k-fold validation, might be not yet statistically robust. More sample and doses should be tested.

Reply. A larger dataset would, indeed, ensure a higher robustness of the model. However, to minimize the variations of the data from tuber to tuber and to focus on the impact of the dose and storage period on the indicator reaction rate, we applied all the doses successively to the samples coming from only two tubers of each variety. Since the experiment was time-sensitive, we could not reliably treat a much larger number of samples than we did (prepare a large number of extracts) and therefore had to work with the available number of samples.

On the other hand, validation is performed anew for each experiment, and this is a robust group validation that confirms the ability of the model to assign unknown samples and eliminates any possibility of overfitting independently of the dataset size. Therefore, there is no reason to doubt the accuracy of the method.

  1. Please explain why did authors just select X-ray irradiation at the doses of 100 and 1000 Gy? Why not 50, or 500 Gy? The storage time is also confusion.

Reply. The dose 100 Gy was selected because it is in the middle of the dose range used for inhibition of potato sprouting. The dose 1000 Gy is known to suppress typical phytopathogens found on the surface of the potato tubers. The monitoring was performed during 6 days because that is when time post-irradiation effects occur in potato tubers, but biochemical and microbiological processes which can impact the dose recognition using fingerprinting technique only start developing. Further, we are going to increase the period of monitoring to bring our experimental findings closer to the real industrial conditions.

  1. The fingerprinting strategy is "black box," what is the method actually sensing? Some information about the chemical changes induced by irradiation that are detected by the indicator reactions should be supplemented.

Reply. The changes in fluorescence intensity during the redox reactions of irradiated potato tubers can be caused by the changing concentrations of a wide variety of compounds the nature of which is not precisely known but can be supposed based on avaliable data. One group is naturally fluorescent compounds, including NADH and flavins. Another group is carbohydrates: due to enzyme activity change by irradiation, starch conversion into sugars can be accelerated. Irradiation can decrease the amount of phenols; their oxidation and the activity of polyphenol oxidase enzyme influence the redox balance, also contributing to the fluorescent fingerprint. Irradiation can increase the formation of reactive oxygen species (ROS) that are highly reactive and can cause oxidative damage to carbohydrates, proteins, lipids, and amino acids; this, in its turn, can create quenchers, fluorescent products or change the indicator reaction rate. The concentration of free amino acids can change during storage and after irradiation due to altered metabolic activity; the amino acids are catalytically active in the indicator reactions, as they can form complexes with metal-ion-catalysts (Cu2+). As a result, combined chemical changes alter the overall metabolic state of the tuber, which may be reflected in the kinetic profile of the indicator reactions. The proposed strategy uses these fluctuations to differentiate between irradiated and non-irradiated potatoes and to estimate the irradiation dose.

After summarizing these possible mechanisms, we became doubtful about the advisability of including this material in the manuscript, as it is primarily speculative in nature, and these assumptions have not been experimentally confirmed specifically for the reaction-based strategy. This approach remains a "black box" technology, where the exact nature of the marker compounds is not precisely known.

  1. How accurate is the present method? Can this method accurately determine the irradiation dose. And it might be largely influenced by the sampling or sample preparation methods. 

Reply. If we understand this question correctly, it is whether we can determine the dose more accurately than to an order of magnitude. – In our studies, we usually irradiated samples at doses differing by an order of magnitude, so the task of determining the dose with greater accuracy could not be solved. This is a challenge for future research. And yes, consistency in sampling and pre-processing methods is crucial; we always ensured that the same protocol was strictly followed at these stages.

Comments on the Quality of English Language

none.

Reviewer 2 Report

Comments and Suggestions for Authors

Hi dear Editorial board and the respected authors

This article "Determining Irradiation Dose in Potato Tubers During Storage using Reaction-based Pattern Recognition Method” has a novelty.

Abstract:

  • The type of statistical design used in this research should be mentioned.
  • Can a wider range of food samples than just raw potato tubers be reliably distinguished between various irradiation doses (such as 0, 100, and 1000 Gy) using the suggested chemical fingerprinting method?
  • What effects do changing storage conditions (such as temperature and humidity) have on the accuracy and effectiveness of the indicator reaction strategy's dose assessment over time?
  • When it comes to identifying low levels of radiation in food products that have been stored, how sensitive are chemometric processing techniques (such as SoftMax regression, LDA, and PLS-DA) compared to conventional instrumental methods?

Introduction:

  • What are the precise mechanisms by which the biochemical composition and shelf-life of potato tubers during extended storage are affected differently by different irradiation doses?
  • In terms of accuracy and dependability for post-irradiation dose assessment in agricultural products, how does the suggested reaction-based fingerprinting strategy stack up against current techniques?
  • What effects might lower-energy irradiation methods have on the effectiveness of pathogen suppression and long-term sprouting prevention in potatoes?

Materials:

  • Please write materials as Company Name (City, Country), especially for chemical analysis assessment which used in the study.

Methodology:

  • What effects does the difference in starch content between the Lina and Agata potato varieties have on their chemical composition and shelf life when exposed to varying radiation doses?
  • What steps were taken to guarantee that the irradiation was consistent throughout the potato samples, and how was the effectiveness of these steps assessed?
  • In what ways do the techniques for measuring fluorescence and absorbance in the redox and aggregation reactions described contribute to the precision and dependability of the fingerprinting analysis for dose assessment?

Results and discussion:

  • What particular chemical alterations in potato tubers cause the fluctuations in fluorescence intensity during the redox reactions over the course of the various storage times?
  • What effects does this have on the stability of the reaction products over time, and how does the time-dependent behavior of the redox reaction signal help distinguish between irradiation doses?
  • What aspects of SoftMax regression outperformed Linear Discriminant Analysis in dose estimation, and how might this affect the selection of analytical methods in subsequent research?

Conclusions:

Conclusion is very general, try to make it more scientific, comprehensive and concise in detail, especially.

References: It is OK.

Comments on the Quality of English Language

The article has many flaws in express and concept of English, it is suggested to be revised in a scientific and native way.

Author Response

Abstract:

  1. The type of statistical design used in this research should be mentioned.

Reply. A completely randomized experimental design was used, which phrase was added to the Abstract.

  1. Can a wider range of food samples than just raw potato tubers be reliably distinguished between various irradiation doses (such as 0, 100, and 1000 Gy) using the suggested chemical fingerprinting method?

Reply. This is a very important question because it determines the possibility of real-world applications of the suggested methodology. The general answer to this question is "Yes," as demonstrated in our previous papers cited in the Introduction (beef, chicken). The low dose (100 Gy) is less confidently discriminated from the control than a 1000 Gy dose. We have added a phrase to the "Conclusions" section, that confirms the applicability of the method to other food products.

  1. What effects do changing storage conditions (such as temperature and humidity) have on the accuracy and effectiveness of the indicator reaction strategy's dose assessment over time?

Reply. In this study, storage conditions were strictly uniform throughout the entire period. There is no indication that temperature or humidity variation in the refrigerator could significantly affect the results of subsequent dose discrimination. However, this could be taken into account in future studies.

  1. When it comes to identifying low levels of radiation in food products that have been stored, how sensitive are chemometric processing techniques (such as SoftMax regression, LDA, and PLS-DA) compared to conventional instrumental methods?

Reply. Conventional techniques vary widely, making direct comparisons with chemometrics-based methods difficult. Sensitivity (as the ability to detect low doses, such as 100 Gy in our studies) varies widely across methods and is insufficient for some of them; in contrast, in our study, 100 Gy was successfully distinguished from the control, albeit with greater effort than 1 kGy. Other existing chemical methods, such as DNA-based tests and antioxidant tests, also target high doses (> 1 kGy). Meanwhile, chemometrics-based methods described in the literature (electronic nose and electronic tongue, [Jo, Y., Ameer, K., Chung, N., Kang, Y. H., Ahn, D. U., & Kwon, J. H. (2020). E-sensing, calibrated PSL, and improved ESR techniques discriminate irradiated fresh grapefruits and lemons. Journal of Food Science and Technology, 57, 364–374. https://doi.org/10.1007/s13197-019-04068-y]) have shown sensitivity to doses < 1 kGy. All of the above applies to foods that have not been stored; for foods that have been stored, the information is too limited. Overall, it is not possible to draw general conclusions about the sensitivity of chemometrics-based dose detection methods compared to others, as this is due to the nature of the method itself.

Introduction:

  1. What are the precise mechanisms by which the biochemical composition and shelf-life of potato tubers during extended storage are affected differently by different irradiation doses?

Reply. Low-dose irradiation (50–150 Gy), that is used for extending the storage life of potatoes primarily by inhibiting sprouting, generates reactive oxygen species, such as hydroxyl radicals, that can cause breaks in the DNA of rapidly dividing cells in the tuber eyes. This mechanism prevents the meristematic tissue from receiving the genetic information needed for cell division and growth, thereby stopping sprouting. Low-dose irradiation can also affect the activity of enzymes involved in carbohydrate metabolism; if altered, it can suppress the sugar accumulation that typically occurs during sprouting [Kumar, S., Bandyopadhyay, N., Saxena, S.; Hajare, S.N.; More, V.; Tripathi, J.; Dahia, Y.; Gautam, S. Differential gene expression in irradiated potato tubers contributed to sprout inhibition and quality retention during a commercial scale storage.  Sci Rep. 2024 14, 13484. https://doi.org/10.1038/s41598-024-58949-0]. Irradiation at doses significantly higher than those needed for sprout inhibition can cause enzyme inactivation, cell wall damage, production of volatile compounds which can produce off-flavors. Doses over 150 Gy can harm the potato's natural immunity, making it more vulnerable to disease and pathogens, which defeats the purpose of extending shelf-life [El-Ramady, H.R.; Domokos-Szabolcsy, É.; Abdalla,   N.A.; Taha, H.S.; Fári, M. Postharvest Management of Fruits and Vegetables Storage. In book: Sustainable Agriculture Reviews; Lichtfouse, E. (Ed.), Springer Cham., 2015, vol. 15, pp. 65–152. https://doi.org/10.1007/978-3-319-09132-7_2].

This text was added to the Introduction.

  1. In terms of accuracy and dependability for post-irradiation dose assessment in agricultural products, how does the suggested reaction-based fingerprinting strategy stack up against current techniques?

Reply. Current methods rely on more or less complex instruments and are highly diverse, applied to different samples under different conditions, and therefore are very difficult to directly compare in performance with ours. Overall, the proposed method does not go beyond discrimination by an order of magnitude, which is weaker than quantitative measurement by existing methods (although high precision may not be practically necessary). The accuracy of instrumental methods is expected to be close to 100%, which is also higher than that of the proposed method (>90% is considered good). Regarding reliability, the signal in the proposed test is based on wet chemical reactions and tends to be less stable over time than signals in instrumental methods. These drawbacks are the price to pay for the simplicity, versatility, and high throughput of the proposed methods. All these features have been analyzed in our previous papers, and we do not believe they should be reiterated in this publication.

Moreover, some of the validated analytical methods fail in identifying the irradiation treatment after a long period of food storage [Zanardi, 2017], which raises interest to the subject of the presented study.

  1. What effects might lower-energy irradiation methods have on the effectiveness of pathogen suppression and long-term sprouting prevention in potatoes?

Reply. This question has been answered above (see i. 5).

Materials:

  1. Please write materials as Company Name (City, Country), especially for chemical analysis assessment which used in the study.

Reply. Materials that were not characterized are now described in Section 2.1.

Methodology:

  1. What effects does the difference in starch content between the Lina and Agata potato varieties have on their chemical composition and shelf life when exposed to varying radiation doses?

Reply. Higher-starch potatoes generally have lower moisture and sugar levels, which makes them potentially more sensitive to starch-to-sugar conversion. Lina is a higher-starch potato and is expected to undergo less dramatic changes than Agata, a lower-starch and higher-moisture variety, which can make it more reactive to irradiation-induced changes. As for the shelf life, the higher starch and lower moisture content in Lina can contribute to a longer shelf life, while the low-starch variety (Agata) may be more susceptible to spoilage (shorter shelf life). This could be exacerbated by radiation due to cell damage and accelerated deterioration. Radiation can cause changes in the starch and sugar content, depending on the temperature and dose, which affects the shelf life. High-starch variety (Lina) is expected to be more stable to irradiation, at the same time being more susceptible to starch-to-sugar conversion when stored in the refrigerator after irradiation. Lower-starch varieties like Agata may be more sensitive to general cellular damage. The overall effect of irradiation depends on the dose and duration of storage. The uncertainty of this situation makes precise forecasting difficult.

  1. What steps were taken to guarantee that the irradiation was consistent throughout the potato samples, and how was the effectiveness of these steps assessed?

Reply. The uniformity of irradiation was addressed in the work in the following way: To ensure uniform irradiation, each side of the potato samples was exposed to irradiation during an equal amount of time (Table 1). Nine identical samples of each variety were irradiated with each dose (Section 2.3). The irradiation method used in the computer simulation was consistent with the irradiation method used in the experiment. Irradiation uniformity in potato samples was simulated using GEANT4 software. As it can be seen from Figure 1,d, the dose uniformity in water parallelepiped was 0.2. However, such dose distribution did not affect the subsequent chemical stage of the experiment, since each of all potato parallelepipeds was used for making the potato extracts (Section 2.4).

Therefore, the uniformity of irradiation was not a concern and could not affect the accuracy of the results.

  1. In what ways do the techniques for measuring fluorescence and absorbance in the redox and aggregation reactions described contribute to the precision and dependability of the fingerprinting analysis for dose assessment?

Reply. In reaction-based optical fingerprinting for dose assessment, the measurements of fluorescence and absorbance contribute significantly to the accuracy and dependability of the overall analysis. Reaction-based fingerprinting relies on monitoring the evolution of a chemical reaction over time, rather than a single static measurement. Absorbance and fluorescence are ideal for this purpose, as they allow for continuous, real-time monitoring of reaction kinetics. The specific rate and pattern of change in the optical signals provide kinetic fingerprints for each sample. By measuring changes in multiple optical parameters, including absorbance and fluorescence at different wavelengths over time, the method generates an information-rich data matrix. This multiparametric approach is more specific and discriminatory than a single measurement, allowing for the differentiation of samples with similar chemical compositions. The different sensitivities of fluorescence and absorbance to chemical changes, particularly in redox and aggregation reactions, provide more unique data for generating a reliable fingerprint. The redundancy of information from multiple measurements makes the overall system more resilient to individual measurement artifacts or noise, thereby increasing the dependability of the final analysis.

Results and discussion:

  1. What particular chemical alterations in potato tubers cause the fluctuations in fluorescence intensity during the redox reactions over the course of the various storage times?

Reply. The fluctuation in fluorescence intensity during the redox reactions of irradiated potato tubers is caused by the changing concentrations of a wide variety of compounds the nature of which is not precisely known but can be supposed based on avaliable data. One group is naturally fluorescent compounds, including NADH and flavins. Another group is carbohydrates: during cold storage, starch in potatoes is converted into sugars, which reaction is accelerated by irradiation; starch can serve as a sorbent preventing compounds that are active in redox reactions from being extracted during sample preparation. Post-irradiation storage can lead to a decrease in phenolic compounds (such as chlorogenic acid); their oxidation and the activity of polyphenol oxidase enzymes influence the redox balance, also contributing to the fluorescent fingerprint. Irradiation and subsequent storage can increase the formation of reactive oxygen species (ROS) that are highly reactive and can cause oxidative damage to carbohydrates, proteins, lipids, and amino acids; this, in its turn, can create quenchers, fluorescent products or change the indicator reaction rate. The concentration of free amino acids can change during storage and after irradiation due to altered metabolic activity; the amino acids are catalytically active in the indicator reactions, as they can form complexes with metal-ion-catalysts (Cu2+). As a result, combined chemical changes alter the overall metabolic state of the tuber, which may be reflected in the kinetic profile of the indicator reactions. The proposed strategy uses these fluctuations to differentiate between irradiated and non-irradiated potatoes and to estimate the irradiation dose.

After summarizing these possible mechanisms, we became doubtful about the advisability of including this material in the manuscript, as it is primarily speculative in nature, and these assumptions have not been experimentally confirmed specifically for the reaction-based strategy. This approach remains a "black box" technology, where the exact nature of the marker compounds is not precisely known.

  1. What effects does this have on the stability of the reaction products over time, and how does the time-dependent behavior of the redox reaction signal help distinguish between irradiation doses?

Reply. First question. As we can understand this question, it is about the stability of irradiation changes in the tubers. – For a short period (a few days), the chemical changes caused by irradiation are stable enough to be reliably observed. However, the metabolic cascade triggered by the treatment make the chemical profile to continue changing during storage, with the rate of change being dependent on the storage temperature. – Second question. The time-dependent behavior of redox reactions acts as a fingerprint because higher irradiation doses generate more reactive compounds that can alter the reaction kinetics. This leads to different kinetic profiles for samples with different doses, allowing for their distinction. The unique shapes of these time-dependent curves are used to discriminate between irradiation doses using chemometric techniques such as LDA, PLS-DA and logistic regression. The algorithm is trained to distinguish between the different dose-dependent patterns. When a sample with an unknown dose is analyzed, its kinetic fingerprint is compared to the established patterns to estimate the dose range.  This material is presented in Sections 2.5 and 3.2–3.5.

  1. What aspects of SoftMax regression outperformed Linear Discriminant Analysis in dose estimation, and how might this affect the selection of analytical methods in subsequent research?

Reply. Softmax regression can outperform LDA on datasets where the classes are not normally distributed or do not have equal variance-covariance matrices. LDA relies on these assumptions, so its performance degrades when they are violated, whereas Softmax regression is more robust to these violations and is better for non-linear decision boundaries. Softmax is also preferred in high-dimensional datasets where LDA covariance matrix can be unstable (added these words to text under Results and discussion). In subsequent studies, for a small number of samples and a large number of variables (many time points and several spectral regions), the SR method may be preferable, while otherwise LDA has better chances of success.

Conclusions:

  1. Conclusion is very general, try to make it more scientific, comprehensive and concise in detail, especially.

Reply. To make the Conclusions more concrete and comprehensive, we added a critical review of the reported technique as follows:

The kinetic-based fingerprinting method that we use is a simple, affordable and rapid technique that does not require full-spectrum instruments or highly qualified personnel. By using other indicator reactions, it can be applied not only to raw potatoes, but also to other food products (beef, chicken [37, 38]). However, these advantages come at the cost of handling a large number of solutions (wet chemistry operations), analyzing standard samples alongside unknowns, and limited precision (up to an order of magnitude). Therefore, the potential for further development of this method lies in the development of ready-to-use test systems ("just add the sample"), improving signal stability over time, and the discrimination of doses that vary slightly from one another.

  1. Thearticle has many flaws in express and concept of English, it is suggested to be revised in a scientific and native way.

Reply. We have carefully checked the manuscript for grammar and vocabulary and believe that now it meets all standards.